# Green Synthesis and Characterization of Silver Nanoparticles of *Psidium guajava* Leaf Extract and Evaluation for Its Antidiabetic Activity

**DOI:** 10.3390/molecules27144336

**Published:** 2022-07-06

**Authors:** Sreeharsha Nagaraja, Syed Sagheer Ahmed, Bharathi D. R., Prakash Goudanavar, Rupesh Kumar M., Santosh Fattepur, Girish Meravanige, Arshia Shariff, Predeepkumar Narayanappa Shiroorkar, Mohammed Habeebuddin, Mallikarjun Telsang

**Affiliations:** 1Department of Pharmaceutical Sciences, College of Clinical Pharmacy, King Faisal University, Al-Ahsa 31982, Saudi Arabia; 2Department of Pharmacology, Sri Adichunchanagiri College of Pharmacy, Adichunchanagiri University, BG Nagara, Mandya 571448, India; sysaha6835@gmail.com (S.S.A.); rambha.eesh@gmail.com (B.D.R.); 3Department of Pharmaceutics, Sri Adichunchanagiri College of Pharmacy, Adichunchanagiri University, BG Nagara, Mandya 571448, India; pgoudanavar01@gmail.com; 4School of Pharmacy, Management and Science University, Seksyen 13, Shah Alam 40100, Malaysia; 5Department of Biomedical Sciences, College of Medicine, King Faisal University, Al-Ahsa 31982, Saudi Arabia; gmeravanige@kfu.edu.sa (G.M.); pshiroorkar@kfu.edu.sa (P.N.S.); hmohammed@kfu.edu.sa (M.H.); 6Department of Pharmaceutics, Vidya Siri College of Pharmacy, Off Sarjapura Road, Bangalore 560035, India; arshia@vidyasiricop.edu.in; 7Department of Surgery, College of Medicine, King Faisal University, Al-Ahsa 31982, Saudi Arabia; mvtelsang@kfu.edu.sa

**Keywords:** *Psidium guajava*, antidiabetic, streptozotocin, silver nanoparticles, characterization

## Abstract

Diabetes mellitus (DM) and its complications are a severe public health concern due to the high incidence, morbidity, and mortality rates. The present study aims to synthesize and characterize silver nanoparticles (AgNPs) using the aqueous leaf extract of *Psidium guajava* (PGE) for investigating its antidiabetic activity. *Psidium guajava* silver nanoparticles (PGAg NPs) were prepared and characterized by various parameters. The in vivo study was conducted using PGE and PGAg NPs in Streptozotocin (STZ)-induced diabetic rats to assess their antidiabetic properties. STZ of 55 mg/kg was injected to induce diabetes. The PGE, PGAg NPs at a dose of 200 and 400 mg/kg and standard drug Metformin (100 mg/kg) were administered daily to diabetic rats for 21 days through the oral route. Blood glucose level, body weight changes, lipid profiles, and histopathology of the rats’ liver and pancreas were examined. In the diabetic rats, PGE and PGAg NPs produced a drastic decrease in the blood glucose level, preventing subsequent weight loss and ameliorating lipid profile parameters. The histopathological findings revealed the improvements in pancreas and liver cells due to the repercussion of PGE and PGAg NPs. A compelling effect was observed in all doses of PGE and PGAg NPs; however, PGAg NPs exhibited a more promising result. Thus, from the results, it is concluded that the synthesized PGAg NPs has potent antidiabetic activity due to its enhanced surface area and smaller particle size of nanoparticles.

## 1. Introduction

Diabetic mellitus is an irreversible disease that is characterized by chronic hyperglycemia caused by dysfunctional lipid, carbohydrate, and protein metabolism [1]. The incidence of type 2 diabetes mellitus (T2DM) is more severe compared to that of type 1 diabetes mellitus (T1DM) and gestational diabetes [2]. The T2DM prevalence is about 90%, affecting approximately 460 million people worldwide, with statistics predicting an increase to over 700 million in 25 years [3]. T1DM results from an autoimmune destruction of beta cells that produce insulin. It is one of the most prevalent pediatric illnesses, with an annual incidence rate of 2% to 5% [4]. T2DM is associated with reduced glucose tolerance due to insulin resistance [5]. Many pharmaceutical medications are utilized to treat diabetes; however, plant-based remedies are often thought to be less toxic and devoid of adverse effects. However, a complex drug molecule causes lower drug absorption, limiting a medicine’s bioavailability. By employing nanotechnology, a drug’s bioavailability can be improved [6,7].

Nanotechnology is one of the most prominent sectors of material science due to its unique physicochemical characteristics. The metal nanoparticles (NP) have enlivened scientific attention, with applications in electronics, catalysis, and drug delivery [8]. NPs are nanomaterials with sizes ranging from 1 to 100 nm. Silver nanoparticles (SNPs) appear under the metal nanoparticles with a broad absorption band in the electromagnetic solar spectrum’s visible zone. SNPs are used in a variety of scientific fields due to their excellent optical characteristics [9]. The advancement of nanobiotechnology has led to the formulation of plant-based SNPs that serve as efficient inhibitors of α-amylase and α-glucosidase enzymes in the management of the most fatal diabetic ailment [10]. The photosynthesized SNPs are inexpensive, environmentally friendly, non-toxic, and safe for human medicinal applications [11]. The plant-based SNPs are proved to inhibit the α-amylase enzyme [12]. Based on this study, in this present communication, the preparation of green synthesis SNPs and evaluation of its antidiabetic property are carried out.

Medicinal plants are in high demand right now and their adaptation is expanding due to their higher availability and lower toxicity to humans [13]. Remarkably, 25% of the modern medications are derived directly or indirectly from plants, proving the strong basis for plant-derived medicine [14,15]. *Psidium guajava,* commonly called guava, is one of the well-known ancient medicinal plants that belongs to the family Myrtaceae. It is available in topical areas like Bangladesh, India, Pakistan, South America, and Indonesia. In many areas, different parts of the guava tree have been utilized to cure diabetes, stomachache, diarrhea, and other health complications. The research studies have shown that *P. guajava* leaves contain numerous phytochemicals that are responsible for a plethora of pharmacological activities, such as antidiabetic, antioxidant, hepatoprotective, antimicrobial, lipid-lowering, and antidiarrheal properties [16].

There is no systematic scientific evidence to prove the antidiabetic activity of PGAg NPs. As a result, the present project aims to evaluate the antidiabetic efficacy of PGAg NPs in STZ-induced diabetic rats.

## 2. Results and Discussion

### 2.1. Phytochemical Constituents Present in the Extracts of P. guajava Leaves

According to preliminary phytochemicals test, *P. guajava* leaf extract contains flavonoids, tannins, phenols, alkaloids, saponins, reducing sugars, proteins, and amino acids but does not contain triterpenoids or glycosides.

### 2.2. Characterization of Psidium Guajava Silver Nanoparticles

#### 2.2.1. UV-Visible Spectroscopy Analysis

The visual observation of color change from pale green to dark brown (Figure 1) is the preliminary step that confirms the formation of AgNPs. This is imputed into the surface Plasmon resonance excitation caused by silver nitrate reduction [17].

The absorption spectrum exhibited a peak at 425 nm (Figure 2), confirming the formation of silver nanoparticles [18].

#### 2.2.2. Fourier Transform Infrared Spectroscopy (FTIR)

According to the FTIR analysis, the PGE contains several functional groups that function as reducing and stabilizing agents prior to formation of NPs. The FTIR spectrum was shown in the region of 4000–500 cm^−1^ (Figure 3). The peak at 2977 cm^−1^ represents the C-H stretching vibration mode in alkanes. The peak at 1582 cm^−1^ represents primary amines (an indication of proteins) in PGE. The peak at 1388 cm^−1^ relates to the O-H bond associated with the characteristic group of phenols. An analysis of the plant extract using FTIR measurements confirmed the presence of active biomolecules that acted as capping and reducing agents, transforming Ag^+^ into Ag^0^ [19].

#### 2.2.3. Scanning Electron Microscopy (SEM)

The SEM study of PGAg NPs revealed the morphological homogeneity in the distribution of Ag NPs on the grid surface. SEM shows an abundance of nanoparticles with a variety of morphologies, though spherical NPs of different sizes tend to predominate. The size of the synthesized Ag NPs ranged between 52.12 nm and 65.02 nm (Figure 4 and Figure 5).

#### 2.2.4. Zeta Potential

The zeta potential of PGAg NPs was observed at −30.7 mV (Figure 6). It confirms that the surface of the nanoparticles is negatively charged. The negative value reflects the particle repulsion and reveals their high stability [20].

### 2.3. Evaluation of Antidiabetic Activity

#### 2.3.1. Blood Glucose Level

In STZ-induced diabetic rats, PGE and PGAg NPs have a significant impact on blood glucose levels, as shown in Table 1. The blood glucose level dropped drastically following PGE, PGAg NPs, and metformin treatment. Both the doses (200 and 400 mg/kg) of PGAg NPs exhibited much higher activity when compared to PGE alone.

#### 2.3.2. Body Weight

In diabetic rats, the loss of body weight was observed from the 1st to 21st day, respectively. Diabetic rats treated with PGE and PGAg NPs experienced considerable body weight recovery as shown in (Table 2).

#### 2.3.3. Biochemical Parameters

On the 21st day after treatment, diabetes rats had elevated total cholesterol (TC), triglycerides (TG), low-density lipoprotein (LDL), and very low-density lipoprotein (VLDL) levels and lower (high-density lipoprotein (HDL) levels compared to normal controls. Such alteration of lipid profile was prevented in the metformin, PGE, and PGAg NPs treated groups. The effect was higher with the 400 mg/kg of PGAg NPs. This shows a significant dose-dependent beneficial effect on lipid profile (Table 3).

#### 2.3.4. Histopathological Examination

##### Liver Cells 

The histoarchitecture of the liver in normal control rats was normal. The hepatic parenchyma and sinusoids seemed normal, with normal Kupffer cell (KC) distribution. The liver of diabetic rats revealed necrotic changes, dilatation of liver sinusoids (LS), activation of KC, and cytoplasmic vacuolization (CV) of hepatocytes. Diabetic rats treated with conventional and test medications showed variable degrees of augmentation. The liver of the metformin-treated group had mild degenerative changes and a moderate hepatic sinusoid dilatation, as well as a slightly larger number of KC. The PG-only treated group’s liver histoarchitecture was more or less normal, with only minimal necrobiotic alterations (NT) and degeneration (FCH). The PGAg NPs group’s liver revealed substantial sinusoidal dilatation, minor necrotic alterations, and a congested CV with the presence of KC (Figure 7).

##### Pancreatic Cells 

The normal control rats’ pancreas exhibited normal histological architecture in the form of acinar structure with normal Langerhans islets. The pancreas of the diabetic control group, in contrast, the number of Langerhans islets (IL) decreased significantly, as well as acini atrophy, vacuolar degeneration, and necroptosis. In both the standard and test medication treated groups, improvements in pancreatic cells were seen. In the metformin-treated group, IL was restored to normal size, with normal acinar cells and little necrotic changes. The PGE-treated group’s pancreas revealed a modest reconstitution of IL cells with minimal necrotic alterations. With the restoration of IL and minimal necrotic alterations, the pancreas of the TPZnO NPs-treated group seemed more or less normal (Figure 8).

## 3. Materials and Methods

### 3.1. Collection of Plant Materials

The leaves of *Psidium guajava* were collected from B.G Nagara, Karnataka, India, and were authenticated by botanists. The leaves were thoroughly washed, shade dried, powdered, and kept at room temperature in an airtight container.

### 3.2. Preparation of Plant Extract

*Psidium guajava* leaves were mixed to double-distilled water in a 1:10 ratio and maintained for 15 min in water bath at 60–70 °C. Then the mixture is super cooled and thoroughly stirred with a magnetic stirrer for 20–30 min for better extraction. Whatman No.1 filter paper was used for removing leaf debris from extract. The obtained clear extract was kept at 4 °C for further use.

### 3.3. Preliminary Phytochemical Screening

The phytochemicals screening of PGE was performed in order to confirm their presence as phytoconstituents [21].

### 3.4. Synthesis of Silver Nanoparticles from P. guajava Extract

For the synthesis, 250 mL of double distilled water was added to a solution of 1 mM silver nitrate (AgNO_3_) and gently vortexed until AgNO_3_ was dissolved. To this solution, we added 5 mL of PGE and mixed well. After 48 h, the change in the hue reflects the development of PGAg NPs. The resultant solution was homogenized for 20 min at 15,000 rpm. The precipitate was collected, and dried NPs were obtained by incubating it at 38 °C [22].

### 3.5. Characterization

The prepared PGAg NPs were subjected to physiochemical characterization by the following methods.

#### 3.5.1. UV-Visible Spectroscopic Analysis

The synthesized silver nanoparticles are characterized by UV-Vis spectroscopy. Using distilled water, silver nanoparticles were diluted to 5 mg/5 mL, and their spectra were measured on a UV-visible spectrometer at 300–800 nm (Shimadzu UV-1800, Shimadzu, CA, USA) [23].

#### 3.5.2. Fourier Transform Infrared Spectroscopy (FTIR)

The synthesized Ag NPs were analyzed to see the organic functional group by FTIR spectroscopy at the region of 4000–500 cm^−1^ (Shimadzu 8400S, Shimadzu, Kyoto Japan).

#### 3.5.3. Scanning Electron Microscopy (SEM) Analysis

The surface morphology of biosynthesized Ag NPs was determined by SEM (ZEISS EVO 18 Research, NY, USA). On a carbon-coated copper grid, the sample was simply dropped on top of the grid to form a thin layer. Blotting paper was used to eliminate the excessive solution and allowed to dry the film under a mercury lamp for up to five minutes [24].

#### 3.5.4. Zeta Potential

To determine the surface charge and stability, the Ag NPs were subjected to photon correlation spectroscopy using Delsa Nano C particle size analyzer [25].

### 3.6. In-Vivo Antidiabetic Activity

#### 3.6.1. Experimental Animals

During the study, both sexes of 160–200 g Wistar rats were employed. A conventional husbandry regimen was followed and fed with standard feed and given free access to water. Animal experimental studies were permitted by the Institutional Animal Ethical Committee (IAEC) of Sri Adichunchanagiri College of Pharmacy (IAEC Approval No. SACCP-IAEC/2020-01/25).

#### 3.6.2. Induction of Diabetes

The rats were intraperitoneally injected with newly prepared STZ (45 mg/kg bw) (Sigma Aldrich, St. Louis, MO, USA) in 0.02 M citrate buffer to induce diabetes. Later, rats were given a 5% dextrose solution to avoid the lethal hypoglycemia caused by STZ. A week after STZ injection, the onset of diabetes was substantiated and included animals possessed fasting plasma glucose levels of >250 mg/dL in this research [26].

#### 3.6.3. Experimental Design

There were seven rat groups, each with six rats. The test samples were administered to all the rats once daily for 21 days. Additionally, 0.1% CMC (Carboxymethylcellulose) in distilled water was used as a suspending agent. As per the literature survey, the *Psidium guajava* extract was found to be safe for up to 5000 mg/kg body weight [27].

**Group I**: Normal control (CMC 0.1%);

**Group II**: Diabetic control received STZ (45 mg/kg b.w.i.p.);

**Group III**: Rats with diabetes treated with Metformin (100 mg/kg b.w.p.o.);

**Group IV**: Rats with diabetes treated with *Psidium guajava* extract (200 mg/kgb.w., p.o);

**Group V**: Rats with diabetes treated with *Psidium guajava* extract (400 mg/kgb.w., p.o);

**Group VI**: Rats with diabetes treated with *Psidium guajava* nanoparticles (200 mg/kgb.w., p.o);

**Group VII**: Rats with diabetes treated with *Psidium guajava* nanoparticles (400 mg/kgb.w., p.o).

#### 3.6.4. Blood Glucose Level

On the 1st, 7th, 14th, and 21st days of treatment, the blood glucose level was measured by the “Rupturing tail vein technique” using a one-touch glucometer (LifeScan, Inc., Malvern, PA, USA).

#### 3.6.5. Assessment of Body Weight

All the experimental rats were weighed from the beginning till the end of the trial at an interval of 7 days (1st, 7th, 14th, and 21st).

#### 3.6.6. Biochemical and Histopathological Analysis

On 21st day of the experiment, the blood was collected using cardiac puncture method under ketamine hydrochloride at 80 mg/kg b.w (i.p.) and centrifuged for 15 min at 3000 rpm and subjected to analyze lipid profiles such as triglycerides (TG), total cholesterol (TC), low-density lipoprotein (LDL), high-density lipoprotein (HDL), and very low-density lipoprotein (VLDL). The liver and pancreas were excised and preserved in a 10% formalin solution for histopathological studies.

### 3.7. Statistical Analysis

Data were expressed as mean ± SEM and differences between the groups were statistically determined by analysis of variance (ANOVA) followed by Dunnett’s test. *p*-values < 0.05 were considered statistically significant.

## 4. Conclusions

The phytochemicals present in the PG extract plays a supreme impact in the nanoparticle formation. The characterization of nanoparticles by UV-visible spectroscopy, FTIR, scanning electron microscopy, and Zeta potential endorse the successful design of silver nanoparticles.

Both extracts and nanoparticles proved a remarkable dose-dependent antihyperglycemic activity, although the nanoparticles surpassed the antidiabetic activity as compared to the extract alone. The biogenic nanoparticles are chemically stable and economically effective. Hence, PGAg NPs may provide an even worthier treatment after being converted into a proper dosage form.

## Figures and Tables

**Figure 1 molecules-27-04336-f001:**
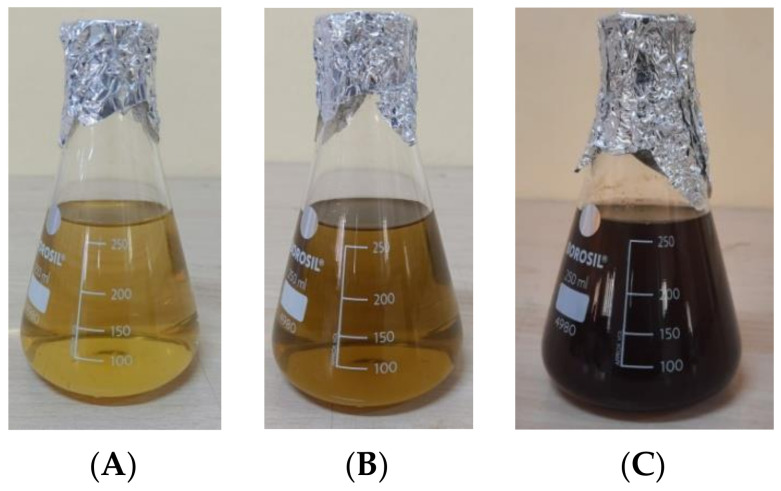
Color changing results in formation of PGAg NPs at (**A**) 0 h, (**B**) 24 h, and (**C**) 48 h.

**Figure 2 molecules-27-04336-f002:**
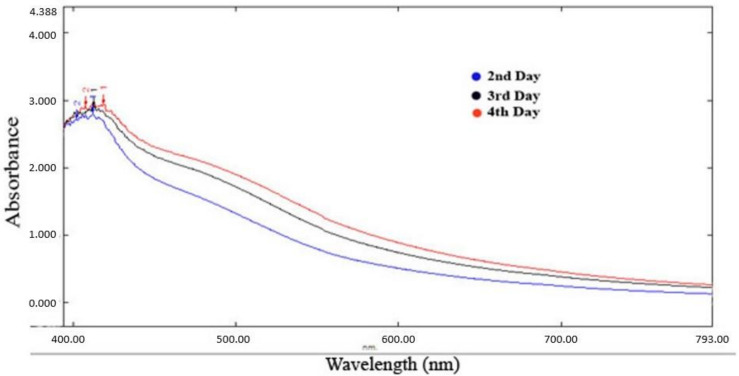
UV-visible spectra Psidium guajava nanoparticles.

**Figure 3 molecules-27-04336-f003:**
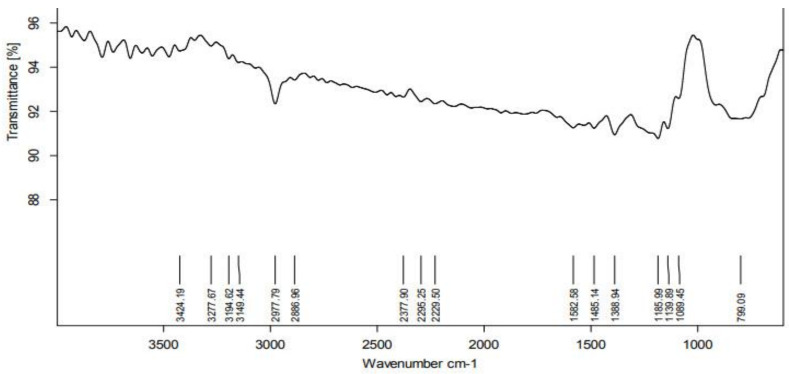
FTIR Spectra of *Psidium guajava* nanoparticles.

**Figure 4 molecules-27-04336-f004:**
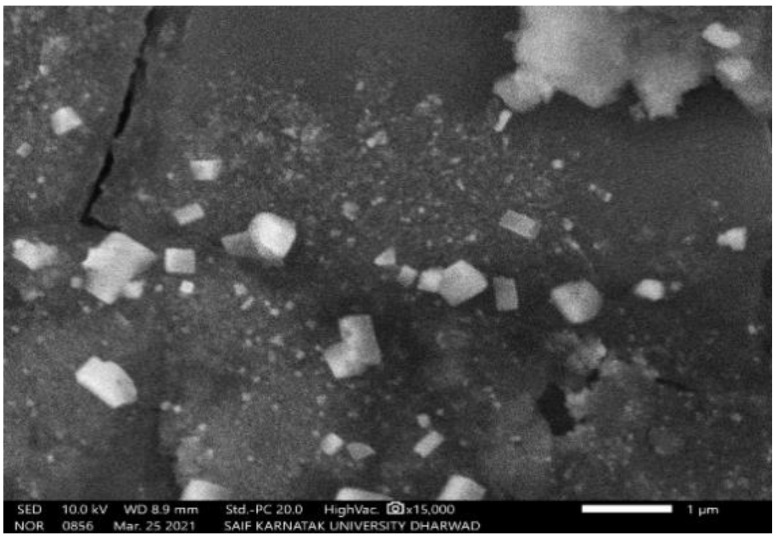
SEM image of PGAg nanoparticles viewed at × 15,000 magnifications with an 8.8 mm scale.

**Figure 5 molecules-27-04336-f005:**
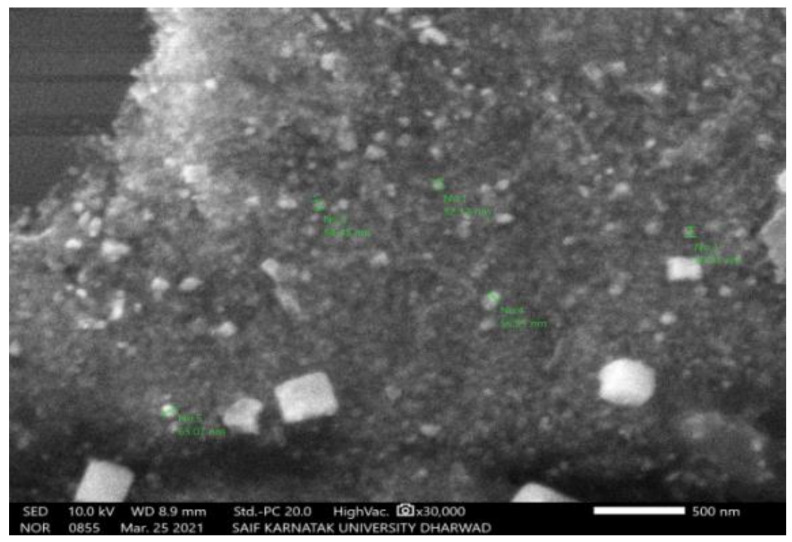
SEM image of PGAg nanoparticles viewed at × 30,000 magnifications with an 8.8 mm scale.

**Figure 6 molecules-27-04336-f006:**
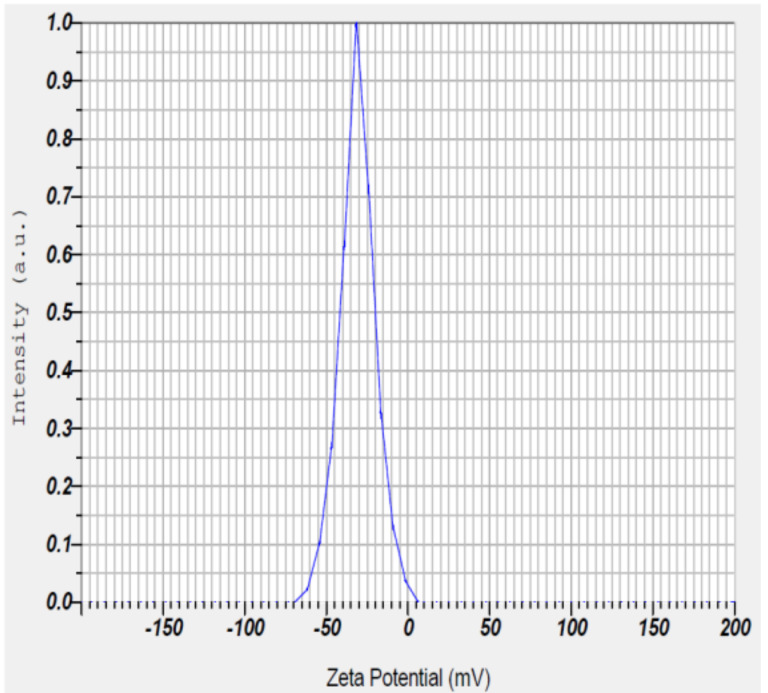
Zeta potential measurement of *Psidium guajava* nanoparticles.

**Figure 7 molecules-27-04336-f007:**
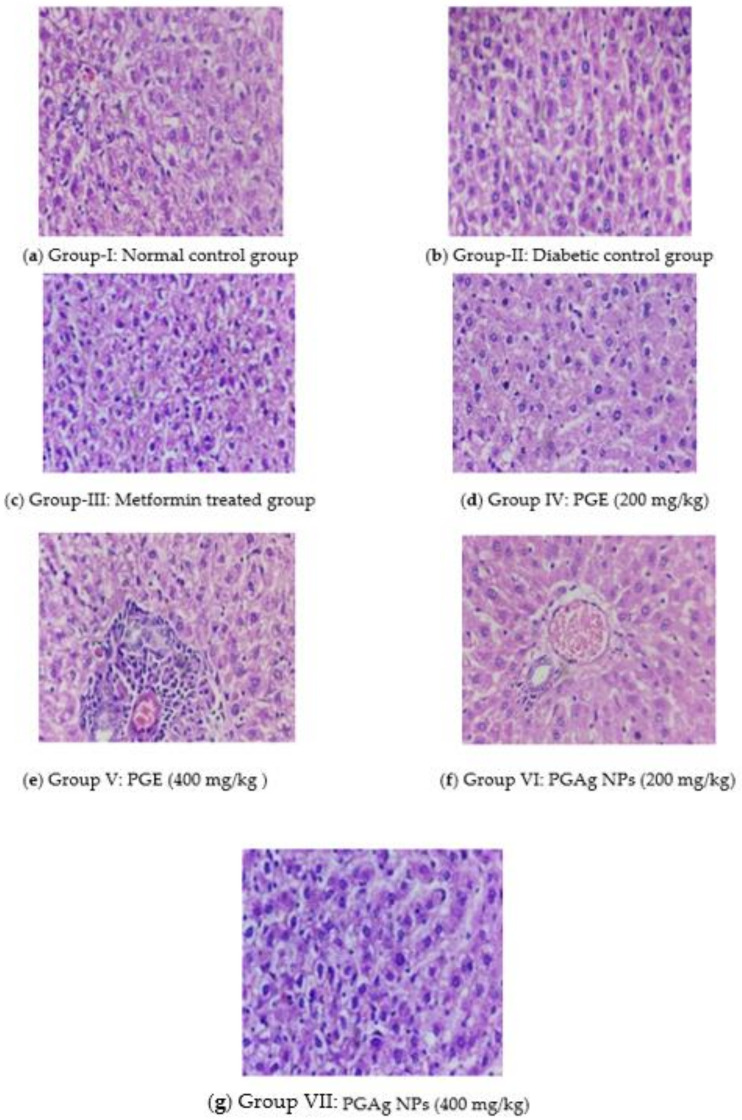
The liver section of histology. (**a**) Group I: normal control group; (**b**) Group II: diabetic control group; (**c**) Group III: metformin-treated group; (**d**) Group IV: PGE (200 mg/kg); (**e**) Group V: PGE (400 mg/kg); (**f**) Group VI: PGAg NPs (200 mg/kg); (**g**) Group VII: PGAg NPs (400 mg/kg).

**Figure 8 molecules-27-04336-f008:**
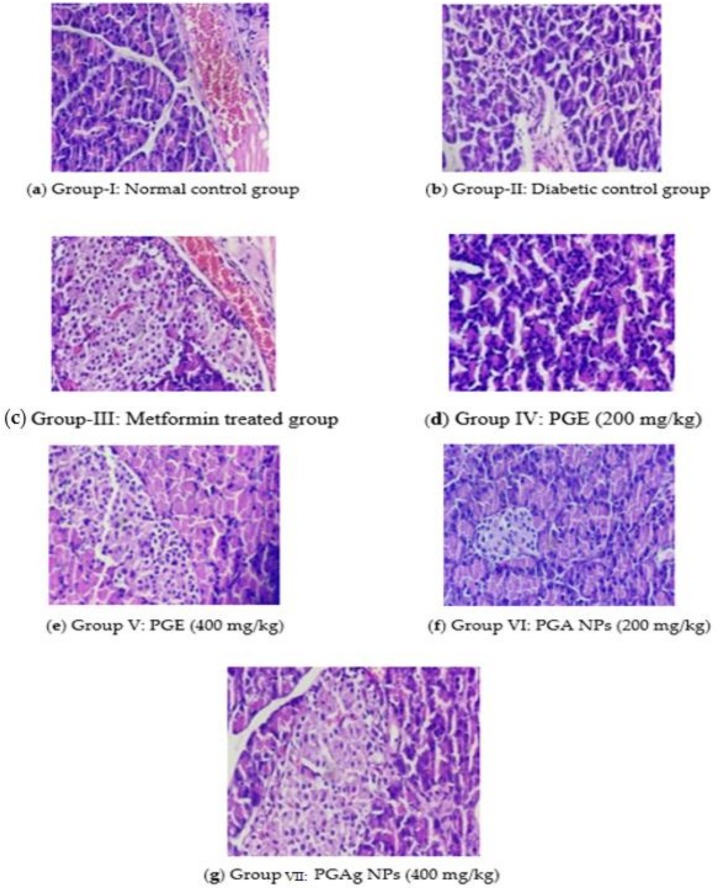
The pancreas section of histology. (**a**) Group I: normal control group; (**b**) Group II: diabetic control group; (**c**) Group III: metformin-treated group; (**d**) Group IV: PGE (200 mg/kg); (**e**) Group V: PGE (400 mg/kg); (**f**) Group VI: PGAg NPs (200 mg/kg); (**g**) Group VII: PGAg NPs (400 mg/kg).

**Table 1 molecules-27-04336-t001:** Effect of *P. guajava* on blood glucose level in STZ-induced diabetic rats.

Group	Blood Glucose Levels (mg/dL) (Mean ± SEM)			
	1st Day	7th Day	14th Day	21st Day
**GP I**	97 ± 0.577	96.5 ± 1.87	95.3 ± 0.88	94.50 ± 0.76
**GP II**	281.16 ± 1.13	289.5 ± 0.76	289.7 ± 0.88	293.3 ± 0.55
**GP III**	231.5 ± 1.25 ^c f^	189.8 ± 0.94 ^c f^	147.5 ± 0.76 ^c f^	104.5 ± 0.76 ^c f^
**GP IV**	251.83 ± 1.3 ^c f^	229.5 ± 0.76 ^c f^	186.5 ± 0.76 ^c f^	135.8 ± 0.6 ^c f^
**GP V**	246.3 ± 1.35 ^c f^	219.7 ± 2.48 ^c f^	161.2 ± 1.07 ^c f^	115.8 ± 1.24 ^c f^
**GP VI**	248.5 ± 1.23 ^c f^	221 ± 1.3 ^c f^	169.8 ± 0.94 ^c f^	126.8 ± 0.79 ^c f^
**GP VII**	243 ± 1.6 ^c f^	207.3 ± 1.52 ^c f^	153 ± 0.81 ^c f^	109.7 ± 0.88 ^c f^

Values are expressed as mean ± SEM, (*n* = 6); *p* < 0.05 ^a^, *p* < 0.01 ^b^, *p* < 0.001 ^c^ compared to diabetic animals and *p* < 0.05 ^d^, *p* < 0.01 ^e^, *p* < 0.001 ^f^ compared to standard drug metformin (two-way ANOVA followed by a Dunnett’s *t*-test). *p*-values < 0.05 were considered statistically significant.

**Table 2 molecules-27-04336-t002:** Effect of PGE and PGAg NPs on body weight (g) in STZ-induced diabetic rats.

Groups	Mean Body Weight (g) (Mean ± SEM)			
	1st Day	7th Day	14th Day	21st Day
**GP I**	189.8 ± 2.89	198.2 ± 3	200.8 ± 2.7	205 ± 2.05
**GP II**	198.2 ± 1.64	180.3 ± 1.66	168 ± 1.5	160 ± 2.0
**GP III**	196 ± 2.7 ^ns^	199.5 ± 2.4 ^c f^	203.2 ± 2.4 ^c f^	206.7 ± 1.92 ^c f^
**GP IV**	191.7 ± 3.45 ^ns^	186.3 ± 3.33 ^ns^	177 ± 3.4 ^a d^	172.1 ± 2.8 ^b e^
**GP V**	194.2 ± 2.38 ^ns^	193.2 ± 2.38 ^b e^	200.8 ± 2.2 ^c f^	206.8 ± 1.77 ^c f^
**GP VI**	197 ± 2.74 ^ns^	191.2 ± 2.3 ^a d^	180 ± 2.1 ^b e^	179.2 ± 1.6 ^c f^
**GP VII**	196.7 ± 3.8 ^ns^	201.5 ± 3.8 ^c f^	205.3 ± 2.72 ^c f^	209 ± 2.38 ^c f^

Values are expressed as mean ± SEM, (*n* = 6); *p* < 0.05 ^a^, *p* < 0.01 ^b^, *p* < 0.001 ^c^ compared to diabetic animals and *p* < 0.05 ^d^, *p* < 0.01 ^e^, *p* < 0.001 ^f^ compared to standard drug metformin (two-way ANOVA followed by a Dunnett’s *t*-test). *p*-values < 0.05 were considered statistically significant.

**Table 3 molecules-27-04336-t003:** Effect of PGE and PGAg NPs on lipid profile in STZ-induced diabetic rats.

Group	Biochemical Parameters(mg/dL)				
	TG	TC	HDL	LDL	VLDL
**GP I**	85.5 ± 0.76	132.5 ± 0.76	53.5 ± 1.17	34.5 ± 0.76	18.5 ± 0.76
**GP II**	143.3 ± 0.88	265.3 ± 0.91	31.33 ± 0.66	65 ± 1.06	41.83 ± 0.79
**GP III**	89.5 ± 0.99 ^c f^	147.7 ± 0.61 ^c f^	45.17 ± 0.94 ^c f^	39.50 ± 0.76 ^c f^	22.5 ± 0.76 ^c f^
**GP IV**	133.8 ± 0.60 ^c f^	252.5 ± 0.76 ^c f^	39.83 ± 0.60 ^c f^	46.17 ± 1.01 ^c f^	35.33 ± 0.88 ^c f^
**GP V**	101 ± 0.856 ^c f^	172.8 ± 0.70 ^c f^	40.17 ± 0.60 ^c f^	43 ± 0.577 ^c f^	26.5 ± 0.76 ^c f^
**GP VI**	134.7 ± 0.76 ^c f^	255.3 ± 0.76 ^c f^	35.5 ± 0.86 ^b e^	55.33 ± 0.88 ^c f^	34.5 ± 0.78 ^c f^
**GP VII**	96.33 ± 0.6 ^c f^	166.2 ± 1.04 ^c f^	42.33 ± 0.61 ^c f^	40.67 ± 0.77 ^c f^	26 ± 0.577 ^c f^

Values are expressed as mean ± SEM, (*n* = 6); *p* < 0.05 ^a^, *p* < 0.01 ^b^, *p* < 0.001 ^c^ compared to diabetic animals and *p* < 0.05 ^d^, *p* < 0.01 ^e^, *p* < 0.001 ^f^ compared to standard drug metformin (two-way ANOVA followed by a Dunnett’s *t*-test). *p*-values < 0.05 were considered statistically significant.

## Data Availability

Data that support the findings of this study are available from the corresponding author upon reasonable request.

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
