# Peer review of "Green Synthesis and Characterization of Silver Nanoparticles of Psidium guajava Leaf Extract and Evaluation for Its Antidiabetic Activity"

_molecules, 2022, doi:10.3390/molecules27144336_

Round 1

Reviewer 1 Report

Dear editor,

Have a nice day. The authors synthesized and characterized silver nanoparticles (AgNPs) for aqueous leaf extract of Psidium guajava (PGE). The antidiabetic activity was evaluated for the synthesized form. The product should a promising antidiabetic activity.

The manuscript needs some minor modifications prior its publication as follows.

1-      Its highly recommended to compare the biological activity of nano form with the extract in the non-nano form.

2-      What is the expected biological target(receptor) of the extract?

3-      Some words should be italic form as in vitro and in vivo.

4-      The toxicity profile of the extract should be assessed.

Author Response

As per reviewer 1 comments, corrections have been made and uploaded.

Reviewer 2 Report

The present study was to synthesize and characterize silver nanoparticles (AgNPs) using aqueous leaf extract of Psidium guajava (PGE) for investigating its antidiabetic activity in rats.

The authors must choose table 2 or figure 8 and table 3 or figure 9.

The experiment is two ways, in table 2 and 3 only show statistical difference between the groups (column) and table not show statistical difference between the time (lines).

The authors must show statistical differences in tables and figures with letters

Edition

Figure 3 erase Bruker.

In table 2 and 3 the authors must write what means the group.

Figure 6. erase the table with the dates of zeta potential and electrophoretic mobility.

Figure 9 What means TG, TC, HDL. LDL and VLDL. 

Conclusion 

Erase 331- The anti-………. research.

Erase 338-340.   This might………effective

Author Response

Response to Reviewer 2 Comments

Point 1-   The authors must choose table 2 or figure 8 and table 3 or figure 9.

Response 1:

As per reviewer 2 comments the table 2 was chosen and fig 8 were removed.

                                                          &

As per reviewer 2 comments the table 3 was chosen and fig 9 were removed.

Point 2-   The experiment is two ways, in table 2 and 3 only show statistical difference between the groups (column) and table not show statistical difference between the time (lines).

Response 2:

Point 3-   The authors must show statistical differences in tables and figures with letters.

Response 3:

Statistical difference has been shown in tables with letters

Point 4-   Edition

Figure 3 erase Bruker.

Response 4:

As per reviewer 2 comments,  in Figure 3,  Bruker was erased.

Point 5-   In table 2 and 3 the authors must write what means the group.

Response 5:

It was included in the table 2 & 3

Point 6-   Figure 6. erase the table with the dates of zeta potential and electrophoretic mobility.

Response 6:

In figure 6, the dates of zeta potential and electrophoretic mobility was removed.

Point 7-   Figure 9 what means TG, TC, HDL. LDL and VLDL. 

Response 7:

The descriptions for TG, TC, HDL. LDL and VLDL. is incorporated. Kindly refer 2.3.3. Biochemical Parameters.

Conclusion 

Point 8: Erase 331- The anti-………. research.

Response 8:

The sentence was removed.

Point 9: Erase 338-340.   This might………effective

Response 9:

The sentence erased.

Reviewer 3 Report

I think this work can be improved. Do you have data with other parts of guava plant based silver nanoparticles? Based on Kumar et al. (2021) work, it shows that guava leaves show high activity but it would be interesting to see how combination of other part extracts with silver nanoparticles will vary their activity. 

Author Response

Response to Reviewer 3 Comments

Point 1-      I think this work can be improved. Do you have data with other parts of guava plant based silver nanoparticles? Based on Kumar et al. (2021) work, it shows that guava leaves show high activity but it would be interesting to see how combination of other part extracts with silver nanoparticles will vary their activity.

Response 1:

All the parts of Psidium guajava may have antidiabetic activity. However, the leaf extract of guava has traditionally been used for the treatment of diabetes in East Asia and other countries. Hence silver nanoparticles of Psidium guajava leaves has been considered for the study.

(Reference article: Mazumdar S, Akter R, Talukder D. Antidiabetic and antidiarrhoeal effects on ethanolic extract of Psidium guajava (L.) Bat. leaves in Wister rats. Asian Pacific Journal of Tropical Biomedicine. 2015 Jan 1;5(1):10-4.

Round 2

Reviewer 2 Report

The authors did the corrections

Reviewer 3 Report

Manuscript looks good in present form and should be accepted for publication. I noticed all the required changes.